# The Skin Sebum and Moisture Levels of Children with Allergic Diseases: How Predictive Are They for House Dust Mite Allergy?

**DOI:** 10.3390/diagnostics14131348

**Published:** 2024-06-25

**Authors:** Seda Çevik, Uğur Altaş, Mehmet Yaşar Özkars

**Affiliations:** Umraniye Training and Research Hospital, Department of Pediatric Allergy and Immunology, University of Health Sciences, Umraniye, 34764 Istanbul, Türkiye; ugur.altas@saglik.gov.tr (U.A.); mehmetyasar.ozkars@sbu.edu.tr (M.Y.Ö.)

**Keywords:** allergy, children, skin sebum, skin moisture, house dust mite

## Abstract

The aim was to evaluate the prediction of house dust mite allergy in children diagnosed with allergic disease based on their skin moisture and sebum levels. This is a case–control study including children with asthma, allergic rhinitis (AR), and atopic dermatitis (AD) and a healthy control group. The participants’ skin moisture and sebum levels were measured non-invasively using a digital device. A total of 421 patients and 143 healthy children were included. The median value of skin moisture percentage was statistically significantly lower in asthma, AR, and AD patients compared to the control group (*p* < 0.001 for each). The median value of skin sebum percentage was significantly lower in asthma and AD patients compared to the control group (*p* = 0.002 and *p* = 0.003, respectively). ROC analysis was performed to assess the predictive value of skin moisture percentage for house dust mite allergy in respiratory allergic diseases (asthma and AR) and AD separately. Using a cut-off point of 35.5% for skin moisture in asthma and AR patients, the sensitivity and specificity were 81.3% and 56.5%, respectively. Although the specificity is low, the high sensitivity value is promising. The non-invasive measurement of skin sebum and moisture could provide convenience to clinicians in the diagnosis and management of allergic diseases.

## 1. Introduction

In recent years, an increase in the frequency of allergic diseases has been observed [1]. Therefore, the prevention and management of allergic diseases are of great importance for child health. Allergic diseases such as asthma, allergic rhinitis (AR), and atopic dermatitis (AD) typically result from an over-reaction of the immune system and can also be associated with environmental factors [2,3]. These environmental factors include exposure to allergens, air pollution, and dietary habits [4,5].

Asthma is one of the most common allergic and chronic diseases in childhood [6]. The most common symptoms of asthma include recurrent attacks of wheezing, coughing, shortness of breath, and chest discomfort [7,8]. Allergic rhinitis is characterized by the inflammation of the mucous membrane inside the nose when exposed to allergens. It manifests with symptoms such as a runny nose, nasal itching, sneezing, and nasal congestion [9]. Atopic dermatitis is the most common non-infectious inflammatory skin disease and exhibits a chronic recurrent nature [10]. The pathophysiology of AD involves a disrupted skin barrier [11,12]. Due to the impaired skin barrier, irritants and allergens can penetrate the skin, initiating the inflammation process associated with AD [13].

The human skin plays important homeostatic roles such as reducing water loss and contributing to thermoregulation [14]. Additionally, our skin serves as the first line of defense against environmental factors [15]. The most important function of the skin is to act as a barrier between the body’s internal and external environments [12]. Maintaining the barrier function of the skin is crucial, as it prevents microorganisms, allergens, and mechanical and chemical irritants from entering the body [12,16].

The presence of sebum and moisture in the skin is necessary for healthy skin. The moisture and sebum balance in the skin affects its barrier function [17]. An impaired skin barrier is the primary abnormality leading to AD [18]. One study in the literature reported that the lipid composition of the skin is different in patients with atopic dermatitis, and these patients exhibit abnormal lipid metabolism in the sebaceous glands [19].

The pathophysiological mechanisms of allergic diseases show similarities. The allergic march refers to the sequential development of asthma, AR, and AD. Although patients with asthma and AR may not have skin findings, allergic diseases can exhibit similar clinical courses and pathophysiological mechanisms, leading to an impaired skin barrier in these patients. There are publications in the literature regarding the low levels of skin moisture and sebum in patients with AD. In a previous study conducted in our clinic with a total of 342 children, including patients with respiratory allergy (asthma and AR) (*n* = 232) and a control group (*n* = 110), we evaluated the skin sebum and moisture content of children without skin findings of dust mite allergy [20]. In this study, we aimed to increase our sample size to make the results more generalizable and representative. Additionally, this study included patients with AD in addition to those with allergic airway diseases. In this context, this study aimed to evaluate the skin moisture and sebum levels in children diagnosed with allergic diseases. Furthermore, this study aimed to determine a cut-off point for these values in predicting the diagnosis of allergen sensitivity. Establishing a cut-off point for non-invasively measured skin sebum and moisture levels in predicting the diagnosis of allergic diseases could facilitate clinicians in clinical practice and guide patients at high risk to allergy clinics.

## 2. Materials and Methods

### 2.1. Study Design, Type, and Sample

This study is a case–control study. Children aged 0–18 years who presented to our pediatric allergy and immunology outpatient clinic with a diagnosis of asthma, AR, and AD were included in the study. Patients with skin diseases other than AD (scabies, seborrheic dermatitis, contact dermatitis, etc.) were excluded from the study. Patients who had taken a bath/shower in the last 24 h, those who had used medications such as local steroids on the day of examination, and those who had used topical products such as moisturizers or additional products (soap, shampoo, detergent, etc.) on the day of examination were excluded from the study due to their potential impact on the skin barrier.

For the study, a control group was formed among children aged 0–18 years who presented to our hospital’s pediatric health and diseases outpatient clinic without a diagnosis of skin disease or any other chronic disease. During the data collection period (approximately 4 months), this study aimed to include all children diagnosed with AR/asthma who visited our pediatric allergy and immunology outpatient clinic and met the inclusion criteria as the patient group. This study aimed to include the maximum number of children in the control group during the data collection period.

### 2.2. Evaluations and Measurement of Skin Sebum and Moisture

The sociodemographic characteristics and skin sebum and moisture values of the patients and the control group were examined. Prior to measurement, children rested for nearly 10–20 min to reduce the effect of physical activity on skin moisture and sebum. At least 1 day before skin moisture and sebum measurement, patients were asked not to use moisturizing cream, sunscreen, or other creams, baby oils, or a coarse scrub in the bath/shower. The skin area to be measured was cleaned with plain water at home on the day of analysis. Measurements were performed in a room with a temperature of approximately 20 °C and air humidity between 40% and 60%. After each measurement, the device was cleaned with an alcohol-containing soft cloth.

Skin moisture and sebum levels were measured using a portable pen-shaped LCD Display Digital Skin Moist Oil Analyzer (Reyoung-Beauty, Shenzhen, China) from the cubital fossa. The device measures moisture and sebum levels using a non-invasive method called bioimpedance. Bioimpedance describes the ability of biological tissue to resist the flow of electric currents, which is a result of the passive electrical properties inherent in biological substances [21]. This device was commercially produced for measuring the moisture and sebum levels of the skin. Measurements were performed on the bare skin for a few seconds by placing the probe of the device on the antecubital fossa of the non-dominant upper limb. The device provides results for the sebum and moisture levels of the skin as percentages. The measurable moisture range is from 0% to 99.9%, and the measurable sebum range is from 16.0% to 63.0%. The device is lightweight, portable, and easy to use, with dimensions of 128 × 26 × 34 mm. While the device has not been validated for diagnostic tests on patients with allergic diseases, previous studies in the literature have utilized it to evaluate skin moisture and sebum content. Additionally, in our study, the eosinophil count, total IgE levels, and the presence of dust mite allergy were evaluated in the patient group. Laboratory values were retrieved from the hospital’s database. The presence of dust mite allergy was determined by having a positive skin prick test (SPT) and/or specific IgE test result.

### 2.3. Statistical Analysis

Data were analyzed with the SPSS (Statistical Package for Social Sciences for Windows 25.0, Armonk, NY, USA) program. Descriptive data were presented with the median, minimum, and maximum values, number (*n*), and percentages (%). The Chi-Square test was used for the comparison of categorized data. The conformity of continuous variables to normal distribution was examined with histograms, probability plots, and Kolmogorov–Smirnov/Shapiro–Wilk tests. The Mann–Whitney U test was used to compare continuous variables that did not conform to normal distribution. The capacity of the skin moisture (%) value in predicting test positivity for house dust mites (skin prick test and/or specific IgE positivity) was analyzed using ROC (Receiver Operating Characteristics) curve analysis. By examining the area under the ROC curve (AUC), we assessed the overall ability of the test to discriminate between positive and negative cases. When a significant cut-off value was observed, the sensitivity and specificity were presented. The ROC curve provides a visual representation of the trade-off between sensitivity and specificity, and the AUC value quantifies the test’s diagnostic accuracy, with a value closer to 1 indicating better performance. The statistical significance level was set at *p* < 0.05.

### 2.4. Ethics

Ethics committee approval was obtained from the Health Sciences University Ümraniye Training and Research Ethics Committee on 5 October 2023 with decision number 356. Before participating in the study, participants and parents were informed about the study, and their consent was obtained.

## 3. Results

In our study, 190 (33.7%) patients with AR, 131 (23.2%) patients with asthma, and 100 (17.7%) patients with AD were included. A total of 143 (25.4%) children were included in the control group. Of the patient group, 53.0% (*n* = 223) were male, and 47.0% (*n* = 198) were female. In the control group, 45.5% (*n* = 65) were male and 54.5% (*n* = 78) were female. There was no statistically significant difference between the patient and control groups in terms of gender and age (*p* > 0.05) (Table 1).

The laboratory parameters of the patients were evaluated. The median values of the absolute eosinophil and eosinophil counts (%) were 260.0 cells/µL (0–2260.0) and 3.2% (0–22.0), respectively, in asthma patients. The median value of total IgE was 99.5 IU/mL (3.0–3840.0). In AR patients, the median values of absolute eosinophils and eosinophils (%) were 300.0 cells/µL (30.0–4160.0) and 3.6% (0.4–21.0), respectively. The median value of total IgE was 177.0 IU/mL (1.0–1802.0). The median values of the absolute eosinophil and eosinophil counts (%) were 430.0 cells/µL (40.0–1840.0), 4.9% (0.5–23.7), respectively, in AD patients. The median value of total IgE was 85.0 IU/mL (2.0–1914.0) (Table 2).

Patients were evaluated for allergens they were sensitive to based on skin prick tests and/or specific IgE levels in the blood. House dust mite allergy was observed in asthma, AR, and AD patients at rates of 42.1% (*n* = 80), 61.1% (*n* = 80), and 54.0% (*n* = 54), respectively (Table 2).

The percentages of the skin moisture and skin sebum of the patients and control group were evaluated. The median value of skin moisture percentage was statistically significantly lower in asthma, AR, and AD patients compared to the control group (*p* < 0.001 for each). The median value of skin moisture percentage was 37.0% (19.0–69.0) in the control group, while it was 35.0% (12.0–56.0), 34.0% (11.0–56.0), and 31.0% (range: 10.0–45.0) in asthma, AR, and AD patients, respectively. The median value of skin sebum percentage was significantly lower in asthma and AD patients compared to the control group (*p* = 0.002 and *p* = 0.003, respectively). Although the median value of skin sebum percentage in AR patients was lower than in the control group, it did not reach statistical significance (*p* = 0.170) (Table 3).

The skin sebum and moisture ratios of patients with asthma, AR, and AD were evaluated according to the presence of house dust mite allergy. In all three patient groups, the skin moisture percentage was lower in those with house dust mite allergen sensitivity compared to those without. This difference was statistically significant in the asthma and AR patient groups (*p* < 0.001 for both). However, in AD patients, there was no significant relationship between the presence of house dust mite allergy and skin moisture percentage (*p* = 0.089). When the relationship between the presence of house dust mites and skin sebum percentages was evaluated, no statistically significant relationship was observed in any of the three patient groups (*p* > 0.005) (Table 4).

ROC analysis was performed to evaluate the predictive value of skin moisture percentage for house dust mite allergy in respiratory allergic diseases (asthma and AR) and AD, separately. Taking 35.5 as the cut-off point for the percent skin moisture of asthma and AR patients, sensitivity and specificity were 81.3% and 56.5%, respectively. The AUC was 71.4% (95.0% CI: 65.8–77.0%) (*p* < 0.001). Taking 31.5 as the cut-off point for the percent skin moisture of AD patients, sensitivity and specificity were 64.8% and 52.2%, respectively. The AUC was 59.8% (95.0% CI: 48.6–71.0%) (*p* = 0.092) (Figure 1).

According to the ROC analysis, the area under curve was low to evaluate the predictive value of skin sebum percentage for house dust mite allergy in respiratory allergic diseases (asthma and AR) and AD.

## 4. Discussion

The prevalence of allergic diseases is increasing, and allergic diseases constitute a significant burden of disease in individuals and society. Therefore, the appropriate and practical management of allergic diseases is of utmost importance. More advanced diagnostic tests and facilities such as skin tests that should be applied for the diagnosis of allergic diseases are generally available in large centers such as tertiary hospitals in our country. In this case, it takes some time for children with allergic symptoms to access an allergy doctor and diagnostic center for diagnosis and evaluation. In some cases, there may even be a loss of follow-up. For this reason, in children whose clinical history and examination suggest allergic diseases and allergen sensitization, skin sebum and moisture measurements with non-invasive methods will make it possible to start the first-line treatment of high-risk children in terms of allergen sensitization until access is gained to a pediatric allergist.

In our study, the median values of skin moisture and skin sebum percentage, which we evaluated with a non-invasive method, were lower in asthma, AR, and AD patients compared to the control group. While this decrease in skin moisture was statistically significant in all three disease groups, the decrease in skin sebum was statistically significant only in asthma and AD patients compared to the control group. In our previous study conducted in our clinic, the skin sebum and moisture content of asthma and AR patients without skin manifestations were found to be significantly lower than the control group [20]. In a study conducted in our country in children with AD, the skin moisture and skin sebum content of children with AD were found to be lower than those of healthy children [22]. The results of this study and the results in the literature suggest that skin moisture and oil balance play an important role in the pathophysiology of AR and asthma as well as AD.

The skin sebum and moisture levels of asthma, AR, and AD patients were evaluated according to the presence of house dust allergen sensitization. In all three patient groups, the percentage of skin moisture was lower in patients with house dust mite allergen sensitization than in those without. This difference was statistically significant in asthma and AR patient groups. There was no statistically significant relationship between house dust mite allergen sensitization and skin sebum. Similar to our previous study, in asthma and AR patients with house dust mite allergen sensitization, the percentages of skin moisture and skin sebum were lower in patients with house dust mite allergen sensitization compared to those without house dust mite allergen sensitization. This decrease in skin moisture was statistically significant [20]. According to the results, it is likely that exposure to inhaled allergens may also affect the skin barrier, which may lead to the loss of skin moisture and a decrease in lipids. According to a study in the literature, it has been reported that even on non-eczematous skin, patients with AD experience impaired barrier function. This impairment was linked to increased sensitization to aeroallergens and could play a role in the development of allergic respiratory symptoms [23]. Further studies are required to elucidate the pathophysiological mechanism underlying the decrease in moisture and sebum content in the skin of asthma and AR patients without skin involvement.

In our study, ROC analysis was performed to evaluate the predictive value of skin moisture percentage for house dust mite allergy both in respiratory allergic diseases and AD. Using a cut-off point of 35.5% for skin moisture percentage, we found a sensitivity of 81.3% and a specificity of 56.5%. Even though the specificity is low, our results indicate that 81.3% of asthma and AR patients with a house dust mite allergy can be identified by measuring skin moisture, which corresponds to approximately four out of five patients—a notably high rate. Similarly, in our previous study, we found a cut-off point of 35.5% for skin moisture percentage in asthma and AR patients, with a sensitivity of 78.9% and a specificity of 57.9% [20]. In our study, in addition to our previous research, we calculated a cut-off point for skin moisture that could predict house dust mite sensitivity in AD patients. However, the sensitivity and specificity values were lower than those in asthma and AR patients (31.5 as the cut-off point, sensitivity and specificity were 64.8% and 52.2%, respectively). Multicenter studies with a higher sample size could be planned to evaluate a cut-off point for skin moisture with high sensitivity and specificity for predicting house dust mite sensitivity in AD patients. Establishing a cut-off point for skin moisture percentage to predict a house dust mite allergy could be a valuable alternative for patients who cannot undergo skin testing or specific IgE level assessment for house dust allergy.

### Limitations and Strengths

There are no similar studies in the literature evaluating the capacity of skin moisture to predict house dust mite allergy in asthma, AR, and AD patients simultaneously. Thanks to our study, it was possible to evaluate skin sebum and moisture in both respiratory allergic diseases and AD with skin involvement and to compare them with the control group. The non-invasive assessment of skin moisture and sebum may shed light on other innovative approaches in the management of allergic diseases. Our study has limitations as well as strengths. Our data collection period was shorter than anticipated, which caused us to not reach the targeted high sample size. This situation creates a limitation in terms of the representation of the population. In addition, the fact that our study was conducted in a single center creates a limitation in terms of the generalizability of the results.

## 5. Conclusions

In our study with asthma, AR, and AD patients, the percentage of skin moisture was significantly lower in the patients than in the control group. Skin moisturization, which is one of the approaches to protect the function of the skin barrier while planning treatment in AD patients, is frequently applied in clinical practice. In patients with respiratory allergic diseases such as asthma and AR, skin involvement is not seen in all patients as it is in atopic dermatitis AD. Therefore, the use of moisturizers in these patients remains less emphasized.

In clinical practice, the measurement of skin moisture and sebum by a non-invasive method in children may be useful in evaluating skin moisture and sebum balance and protecting the barrier function of the skin. Since the presence of allergen sensitization such as house dust mites may be possible in values below 35.5%, which is the cut-off value we found for skin moisture in asthma and AR, our study results will be guiding in terms of referring these children to pediatric allergy clinics. There are studies in the literature regarding predictors of allergic situations [24,25]. Our study also contributes to this field in the literature. In addition, our study results may shed light on the adoption of holistic and innovative approaches in the treatment and management of allergic diseases.

## Figures and Tables

**Figure 1 diagnostics-14-01348-f001:**
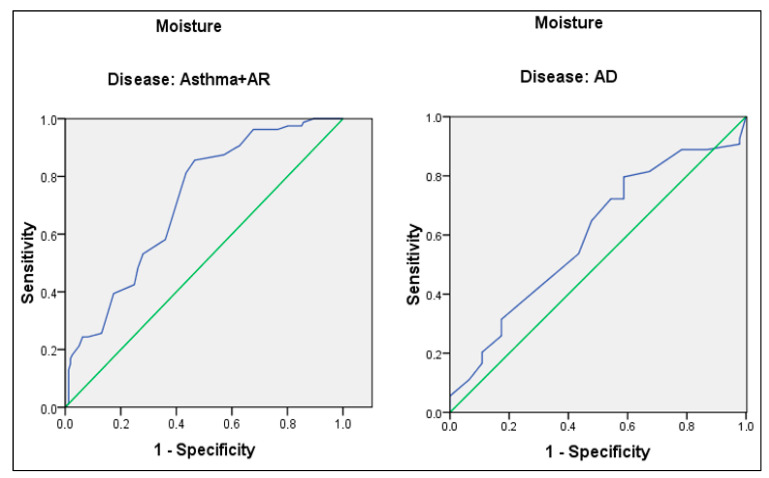
ROC curve for skin moisture content of asthma, AR, and AD patients. Blue line: ROC curve, Green line: Diagonal reference line.

**Table 1 diagnostics-14-01348-t001:** The age and gender of the patients and control group.

	Asthma	AR	AD	Control Group	*p* Value
Gender, *n* (%)					
Female	84 (44.2)	60 (45.8)	54 (54.0)	78 (54.5)	0.120
Male	106 (55.8)	71 (54.2)	46 (46.0)	65 (45.5)	
	Total: 198 (47.0) 223 (53.0)				
Age (years), median (min–max)	6 (1–17)	6 (0–17)	4 (0–16)	7 (0–17)	0.525
	Total: 6 (0–17)				

AD: atopic dermatitis, AR: allergic rhinitis; *p* values were calculated for total patient group and control group.

**Table 2 diagnostics-14-01348-t002:** The eosinophil counts, total IgE levels, and allergy test positivity of the patients.

	Asthma	AR	AD
Laboratory Parameters	Median (Min–Max)	Median (Min–Max)	Median (Min–Max)
Eosinophil (absolute) (cells/µL)	260.0 (0–2260.0)	300.0 (30.0–4160.0)	430.0 (40.0–1840.0)
Eosinophil (%)	3.2 (0–22.0)	3.6 (0.4–21.0)	4.9 (0.5–23.7)
Total IgE (IU/mL)	99.5 (3.0–3840.0)	177.0 (1.0–1802.0)	85.0 (2.0–1914.0)
	*N* (%)	*N* (%)	*N* (%)
Allergen sensitivity for house dust mite	80 (42.1)	80 (61.1)	54.0 (54.0)

AD: atopic dermatitis, AR: allergic rhinitis.

**Table 3 diagnostics-14-01348-t003:** Moisture and sebum of patient and control group.

	Asthma	AR	AD	Control Group
	Median (Min–Max)	Median (Min–Max)	Median (Min–Max)	Median (Min–Max)
Moisture	35.0 (12.0–56.0)	34.0 (11.0–56.0)	31.0 (10.0–45.0)	37.0 (19.0–69.0)
*p* value	<0.001	<0.001	<0.001	
Sebum	25.0 (16.0–49.0)	25.0 (15.0–55.0)	25.0 (16.0–49.0)	28.0 (16.0–51.0)
*p* value	0.002	0.170	0.003	

*p* values were calculated for asthma versus control group, AR versus control group, and AD versus control group. AD: atopic dermatitis, AR: allergic rhinitis.

**Table 4 diagnostics-14-01348-t004:** Moisture and sebum content of asthma, AR, and AD patients with and without house dust mite allergy.

				Median	Minimum	Maximum	* *p* Value	** *p* Value
Asthma	House dust mite allergy	No	Moisture	37.0	25.0	56.0	<0.001	0.637
			Sebum	25.5	16.0	49.0		
		Yes	Moisture	34.0	12.0	50.0		
			Sebum	24.0	16.0	45.0		
AR	House dust mite allergy	No	Moisture	36.0	11.0	56.0	<0.001	0.292
			Sebum	25.0	16.0	49.0		
		Yes	Moisture	32.0	11.0	49.0		
			Sebum	25.5	15.0	55.0		
AD	House dust mite allergy	No	Moisture	32.0	12.0	45.0	0.089	0.605
			Sebum	24.0	18.0	49.0		
		Yes	Moisture	30.0	10.0	45.0		
			Sebum	25.0	16.0	49.0		

AD: atopic dermatitis, AR: allergic rhinitis. * *p* value was calculated for moisture content of patients with and without house dust mite allergy. ** *p* value was calculated for sebum content of patients with and without house dust mite allergy.

## Data Availability

Data are contained within the article.

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
