# Peer review of "The Skin Sebum and Moisture Levels of Children with Allergic Diseases: How Predictive Are They for House Dust Mite Allergy?"

_diagnostics, 2024, doi:10.3390/diagnostics14131348_

Round 1
Reviewer 1 Report
Comments and Suggestions for Authors
This is a case-control study including children with asthma, allergic rhinitis (AR),
and atopic dermatitis (AD) and a healthy control group. The skin moisture and sebum levels
were measured non-invasively method. The study was divided into 2 groups: Patients and healthy control. This study had determined a cutoff point on the percentage of skin moisture which can be used to determine the allergenicity of a patient in diagnosis and management of allergic diseases.
This study is a continuation of previous research reported by the same group of authors.
Comments:
1. Inconsistency of the abbreviation is noticed, i.e. sometimes allergic rhinitis was used, sometimes AR was used.
2. Authors need to provide the model and details of the device used in the measurement.
3. Authors did not described how eosinophil count, total IgE levels and the dust mite allergy were sampled and analysed.
4. The data in Table 1 was not aligned properly.
5. The point (.) should be represented as “.”, not comma (,). i.e. it should be 35.0, not 35,0.
6. Authors need to explain how does the ROC analysis is been conducted.
7. The last sentence in conclusion “In patients with asthma and AR, which are…..”, this sentence is unclear. The authors need to explain this further.
Author Response
Point 1: Inconsistency of the abbreviation is noticed, i.e. sometimes allergic rhinitis was used, sometimes AR was used.
Answer 1: Dear Reviewer, thank you for your suggestions. We have corrected the inconsistencies.
Point 2: Authors need to provide the model and details of the device used in the measurement.
Answer 2: Thank you for your suggestion. We have added the information about the device. You can find it in “Evaluations and Measurement of Skin Sebum and Moisture” of the methods part in our revised manuscript.
Point 3: Authors did not described how eosinophil count, total IgE levels and the dust mite allergy were sampled and analysed.
Answer 3: Thank you for your comment. The information you mentioned has been added to the revised manuscript as follows: “We have conducted this study retrospectively, so the laboratory values were retrieved from the hospital’s database. The presence of dust mite allergy was determined by having a positive skin prick test (SPT) and/or specific IgE test result.”
Point 4: The data in Table 1 was not aligned properly.
Answer 4: We have corrected it.
Point 5: The point (.) should be represented as “.”, not comma (,). i.e. it should be 35.0, not 35,0.
Answer 5: We have corrected it.
Point 6: Authors need to explain how does the ROC analysis is been conducted.
Answer 6: We have added the information to statistical analysis part :“ By examining the area under the ROC curve (AUC), we assessed the overall ability of the test to discriminate between positive and negative cases.When a significant cutoff value was observed, the sensitivity and specificity were presented. The ROC curve provides a visual representation of the trade-off between sensitivity and specificity, and the AUC value quantifies the test's diagnostic accuracy, with a value closer to 1 indicating better performance.”
Point 7: The last sentence in conclusion “In patients with asthma and AR, which are…..”, this sentence is unclear. The authors need to explain this further.
Answer 7: We revised the sentence that you mentioned as following: “In patients with respiratory allergic diseases such as asthma and AR, skin involvement is not seen in all patients as it is in atopic dermatitis AD. Therefore, the use of moisturizers in these patients remains less emphasized.”
Reviewer 2 Report
Comments and Suggestions for Authors
In the manuscript titled: “The Skin Sebum and Moisture Levels of Children with Allergic Diseases: How Predictive Are They for House Dust Mite Allergy?”, the authors evaluated the prediction of house dust mite allergy in children diagnosed with allergic disease based on their skin moisture and sebum levels using a case-control study including children with asthma, allergic rhinitis (AR), and atopic dermatitis (AD) and a healthy control group.
The study is interesting and the general logic of the article is fairly integral. The presentations are clear. However, certain statements require further clarification such as “applying our clinic” in the methodology section
Comments on the Quality of English LanguageIn the manuscript titled: “The Skin Sebum and Moisture Levels of Children with Allergic Diseases: How Predictive Are They for House Dust Mite Allergy?”, the authors evaluated the prediction of house dust mite allergy in children diagnosed with allergic disease based on their skin moisture and sebum levels using a case-control study including children with asthma, allergic rhinitis (AR), and atopic dermatitis (AD) and a healthy control group.
The study is interesting and the general logic of the article is fairly integral. The presentations are clear. However, certain statements require further clarification such as “applying our clinic” in the methodology section
Author Response
Point 1: The study is interesting and the general logic of the article is fairly integral. The presentations are clear. However, certain statements require further clarification such as “applying our clinic” in the methodology section
Answer 1: Dear Reviewer, thank you for your suggestion. We revised the sentence that you mentioned as following: “..it was aimed to to include all children diagnosed with AR/asthma who visited our Pedi-atric Allergy and Immunology outpatient clinic and met the inclusion criteria as the pa-tient group”